# Novel Class of Chalcone Oxime Ethers as Potent Monoamine Oxidase-B and Acetylcholinesterase Inhibitors

**DOI:** 10.3390/molecules25102356

**Published:** 2020-05-18

**Authors:** Jong Min Oh, T. M. Rangarajan, Reeta Chaudhary, Rishi Pal Singh, Manjula Singh, Raj Pal Singh, Anna Rita Tondo, Nicola Gambacorta, Orazio Nicolotti, Bijo Mathew, Hoon Kim

**Affiliations:** 1Department of Pharmacy, and Research Institute of Life Pharmaceutical Sciences, Sunchon National University, Suncheon 57922, Korea; ddazzo005@naver.com; 2Department of Chemistry, Sri Venketeswara College, University of Delhi, New Delhi-110021, India; rpsingh54@gmail.com; 3Centre for Fire, Explosive and Environment Saftey, DRDO, Delhi-110054, India; chaudhary.reeta90@gmail.com (R.C.); rajcfees@gmail.com (R.P.S.); 4Department of Chemistry, University of Dehli, Dehli-110007, India; 5Department of Chemistry, Shivaji College, University of Delhi, New Delhi-110027, India; manjulasingh56@gmail.com; 6Dipartimento di Farmacia—Scienze del Farmaco, Università degli Studi di Bari “Aldo Moro”, via E. Orabona, 4, I-70125 Bari, Italy; annarita.tondo@gmail.com (A.R.T.); nicola.gambacorta1@uniba.it (N.G.); onicolotti@gmail.com (O.N.); 7Division of Drug Design and Medicinal Chemistry Research Lab, Department of Pharmaceutical Chemistry, Ahalia School of Pharmacy, Palakkad-678557, Kerala, India

**Keywords:** monoamine oxidase, acetylcholinesterase, chalcones, oximes, kinetics, reversibility

## Abstract

Previously synthesized novel chalcone oxime ethers (COEs) were evaluated for inhibitory activities against monoamine oxidases (MAOs) and acetylcholinesterase (AChE). Twenty-two of the 24 COEs synthesized, except **COE-17** and **COE-24**, had potent and/or significant selective inhibitory effects on MAO-B. **COE-6** potently inhibited MAO-B with an IC_50_ value of 0.018 µM, which was 105, 2.3, and 1.1 times more potent than clorgyline, lazabemide, and pargyline (reference drugs), respectively. **COE-7**, and **COE-22** were also active against MAO-B, both had an IC_50_ value of 0.028 µM, which was 67 and 1.5 times lower than those of clorgyline and lazabemide, respectively. Most of the COEs exhibited weak inhibitory effects on MAO-A and AChE. **COE-13** most potently inhibited MAO-A (IC_50_ = 0.88 µM) and also significantly inhibited MAO-B (IC_50_ = 0.13 µM), and it could be considered as a potential nonselective MAO inhibitor. **COE-19** and **COE-22** inhibited AChE with IC_50_ values of 5.35 and 4.39 µM, respectively. The selectivity index (SI) of **COE-22** for MAO-B was higher than that of **COE-6** (SI = 778.6 vs. 222.2), but the IC_50_ value (0.028 µM) was slightly lower than that of **COE-6** (0.018 µM). In reversibility experiments, inhibitions of MAO-B by **COE-6** and **COE-22** were recovered to the levels of reference reversible inhibitors and both competitively inhibited MAO-B, with K_i_ values of 0.0075 and 0.010 µM, respectively. Our results show that **COE-6** and **COE-22** are potent, selective MAO-B inhibitors, and **COE-22** is a candidate of dual-targeting molecule for MAO-B and AChE.

## 1. Introduction

Alzheimer’s disease (AD) is one of the greatest concerns confronting the medical community, and is the fourth leading cause of neurodegenerative disease-related death. Furthermore, AD has been predicted to affect 100 million patients within 30 years [1]. AD accounts for 70% of all reported cases of dementia, being characterized by cholinergic functional decline, β-amyloid oligomer formation, and the dysregulations of other cellular processes [2]. Over past years, many efforts have been made to identify the key biochemical events responsible for AD. However, AD is a multifactorial disease and, thus, its management requires the simultaneous modulations of multiple targets [3]. Based on greater understanding of the disease, recent research efforts have increasingly focused on multitarget-drugs that simultaneously bias different biological targets [4].

This novel approach is viewed optimistically, and hybridizations of the pharmacophore subunits of bioactive molecules have already resulted in the identification of multifunctional drugs [5] and, as a result, synthetic drugs, like donepezil, rivastigmine, and tacrine, have been used as structural models for molecular hybridization experiments (Figure 1) [6]. Tacrine was the first cholinesterase (ChE) inhibitor that was approved by the FDA for the treatment of AD. However, the use of tacrine is limited by its side-effects and, thus, searches for more compatible and potent tacrine derivatives continue [7].

On the other hand, monoamine oxidase (MAO)-A is primarily targeted for the treatment of depression and anxiety, whereas MAO-B is targeted for AD and Parkinson’s disease, based on their specificity, which is, MAO-A prefers serotonin, and MAO-B prefers phenylethylamine and benzylamine [8]. Rasagiline is a MAO inhibitor, and its neuroprotective activity has been attributed to the presence of a propargyl amine moiety, which suppresses the overexpression of Bax protein in AD [9].

The complexity of AD militates against the use of consolidated mono-therapies and supports the notion that dual MAO and acetylcholinesterase (AChE)-inhibitory activities are likely to have better therapeutic effects in AD [10]. Ladostigil is an example of such multi-functional drugs, as it possesses the neuroprotective effects of rasagiline and ChE inhibitory activity (Figure 1) [11]. Notably, most drugs used to treat AD patients in palliative care settings are ChE inhibitors with some multifunctional activity. Furthermore, many studies have shown that MAO inhibitors have attracted considerable research interest in the context of halting or retarding the progression of AD [4].

Chalcones are versatile scaffolds and they are widely distributed in edible plants. Several attempts have been made to synthesize novel biologically active chalcone derivatives due to their wide-ranging biological activities [12,13,14,15,16]. Over recent decades, the MAO-B inhibitory activities of chalcone derivatives have progressively appreciated [17], and many studies have reinforced the association between chalcone derivatives and potent MAO-B inhibition [18,19,20,21,22,23,24,25,26,27,28,29,30,31,32,33,34,35]. Recently, our group reported that ethoxy and ethyl acetohydroxamate (Figure 2) functionalities on chalcone phenyl groups confer significant MAO-B and AChE inhibitory effects [36,37].

The oxime ethers are among the most important structural pharmaceutical motifs, for example, they have been associated with transthyretin amyloid fibril formation [38], antibiotic (Cefmenoxime [39], Aztreonam [40], Roxithromycin [41]), anti-inflammatory (Ridogrel) [42], antifungal (Oxiconazole) [43] (Figure 2), and neuroleptic activities [44]. We considered manipulating the ethyl acetohydroxamate functionality in ethyl acetohydroxamate chalcones by introducing an oxime to produce a range of novel chalcone oxime ethers (COEs) with the objective of synthesizing drugs with MAO and AChE inhibitory effects for the treatment of AD since oxime ethers have numerous biological properties and ethyl acetohydroxamate chalcones have significant MAO-B and AChE inhibitory [37] (Figure 2) and antiplasmodial [45] effects. Recently, we used Pd-catalyzed C‒O cross-coupling reactions between bromo-chalcones and aldoximes [46], or ketoximes [47] in order to synthezise the chalcones, as shown in Scheme 1.

Herein, we report the abilities of our previously synthesized chalcone ketoxime ethers to inhibit human MAOs (hMAOs) and AChE, kinetics, reversibility, and docking studies.

## 2. Results and Discussion

### 2.1. Synthesis of COEs

Pd-catalyzed C‒O cross-coupling was used to produce 24 COEs by reacting activated aryl bromides, ketoximes, and chalcone oximes together, as shown in Scheme 2, and previously described [47].

The activated aryl bromides included aryl bromides bearing electron-withdrawing groups at the 4-position and bromo-chalcones. Screening phosphine ligands, Pd-catalyst, and solvents was utilized to optimize the method.

### 2.2. Inhibitory Activities against MAO-A, MAO-B and AChE

The MAO-A, MAO-B, and AChE inhibitory activities of 24 synthesized COEs were evaluated while using toloxatone, lazabemide, clorgyline, pargyline, and tacrine as reference molecules (Table 1). The synthesized COEs were of two structural categories, that is, chalcone ketoxime or chalcone-chalcone oxime hybrids. Nineteen of the 24 COEs showed residual MAO-B activities of <50% at 1.0 µM and potently inhibited MAO-B with IC_50_ values of <1.0 µM (Table 1). Eight of the 19 showed significant MAO-B inhibitory activities with IC_50_ values of <0.1 µM, and compound **COE-6** most potently inhibited MAO-B (IC_50_ = 0.018 µM), and **COE-7** and **COE-22** had equally significant MAO-B inhibitory activity (both IC_50_ = 0.028 µM). Interestingly, **COE-6** and pargyline had an identical IC_50_ value (0.020 µM) and they were 2.33 and 105.6 times more potent than lazabemide and clorgyline, respectively, and **COE-6** was more potent than other chalcone derivatives, (2E)-1-(4-ethoxyphenyl)-3-(4-fluorophenyl) prop-2-en-1-one (**E7,** IC_50_ = 0.053 µM) [36], and ethyl (1E)-N-{4-[(1E)-3-(4-fluorophenyl)-3-oxoprop-1-en-1-yl]phenoxy}ethanimidate (**L3**, IC_50_ = 0.053 µM) [37]. Similarly, **COE-7** and **COE-22** were 1.5 and 68 times more potent than lazabemide and clorgyline, respectively.

Twenty three of the 24 COEs showed residual MAO-A activities of >60% at 1.0 µM, but only **COE-13** had a residual activity of <50% at 1.0 µM (Table 1). Twenty-three of the COEs screened relatively weakly inhibited MAO-A (IC_50_ > 4.0 µM). **COE-13** had an IC_50_ value of 0.88 µM and it was 1.1 and 2.8 times more potent than toloxatone (IC_50_ = 0.99 µM) and pargyline (IC_50_ = 2.43 µM), respectively. **COE-13** also significantly inhibited MAO-B (IC_50_ = 0.13 µM) and, thus, had a low selectivity index (SI) of 6.8. Twenty two of the 24 COEs inhibited MAO-B more than MAO-A, as was reflected by SI values (defined as the ratio of the IC_50_ values of MAO-A to MAO-B). Of the 24 COEs, only **COE-6**, **COE-7**, **COE-8**, **COE-21**, and **COE-22**, had high SI values. Compound **COE-8** (IC_50_ = 0.042 µM) had the lowest SI (182.9), and the most potent MAO-B inhibitor **COE-6** (IC_50_ = 0.018 µM) had the second lowest (222.2). The next most potent MAO-B inhibitors **COE-7** and **COE-22** (both had an IC_50_ of 0.028 µM) had SI values of 392.9 and 778.6, respectively, and **COE-21** (IC_50_ = 0.036 µM) had the second highest SI (441.7). This result shows that the chalcone-chalcone oxime hybrids tend to inhibit MAO-B more selectively than chalcone ketoxime hybrids.

AChE inhibition studies showed that only six compounds, **COE-5**, **COE-9**, **COE-10**, **COE-19**, **COE-21**, and **COE-22**, had AChE residual activities of <50% at 10 µM with IC_50_ values of 7.06, 8.39, 9.42, 5.35, 9.65, and 4.39 µM, respectively. However, AChE inhibitions by these compounds were approximately 22 times less than that of the reference tacrine. **COE-22** most potently inhibited AChE and also inhibited MAO-B well and it had the highest SI value, which suggest its possible use for the dual-targeting of MAO-B and AChE.

### 2.3. SARs for Inhibition Studies

Twenty two of 24 COEs included in the present study were selective MAO-B inhibitors (Table 1). The activity results showed that the activity of the COEs depended on the structures and substituents of oximes and chalcones. For example, in compounds **COE-1** to **COE-4**, structures and the substituent (-OMe) on the chalcone moiety were the same and substituents and structures of oximes differed. The substituents on acetophenone oxime of the chalcone moieties of **COE-1** and **COE-2** had no meaningful influence on MAO-B inhibition, whereas the conversion of acetophenone oxime (**COE-1**) to benzophenone oxime (**COE-3**) and 1-indanone oxime (**COE-4**) on the chalcone moiety increased MAO-B inhibition by 5.8 and 4.7 times, respectively. Similarly, the removal of –OMe from chalcone and introduction of a F at the 4-position of acetophenone oxime of **COE-1** greatly reduced the MAO-B inhibitory activity of **COE-5** (IC_50_ = 1.12 µM) four-fold. Introduction of F in place of the -OMe group in chalcone moiety of **COE-3** structure (IC_50_ = 0.048 µM) enhanced the MAO-B inhibitory activity of the compound **COE-7** (IC_50_ = 0.028 µM) 1.7-fold, which suggested that the presence of F increases MAO-B inhibitory activity more than –OMe group, similar to other chalcone derivatives containing F [48]. Similarly, replacement of the benzophenone oxime of **COE-7** with cyclohexanone oxime in **COE-6** (IC_50_ = 0.028 µM) enhanced MAO-B inhibitory activity by 1.5-fold. The presence of a F in acetophenone oxime in **COE-5** (IC_50_ = 1.12 µM) and in **COE-11** (IC_50_ = 1.82 µM) reduced MAO-B inhibition, irrespective of the substituents on or the position of oxime in chalcone structure. The position of the 1-acetonaphthone oxime in the chalcone structures of **COE-8** (IC_50_ = 0.042 µM) and **COE-14** (IC_50_ = 1.58 µM) also reduced MAO-B inhibitory activity. Similarly, the position of benzophenone oxime in the chalcone structures of **COE-3** (IC_50_ = 0.048 µM) and **COE-15** (IC_50_ = 0.95 µM) significantly altered MAO-B inhibitory activity. As a result, different ketoxime structures in the benzaldehyde portion of the chalcone structure had significantly greater MAO-B inhibitory activities than ketoxime structures in the acetophenone portion of chalcones. Our SAR study suggested that active oxime groups (e.g., benzophenone oxime, cyclohexanone oxime, acetothiophene oxime, and 1-acetonaphthone oxime) on benzaldehyde portions of chalcones and altering substituents and the structure on the acetophenones portion of chalcones provide a means of enhancing MAO-B inhibitory activities.

Of the three *O*-aryl chalcone oximes (**COE-16**, **COE-17**, and **COE-18**), two compounds, **COE-17** and **COE-18**, showed significantly higher MAO-B inhibitory activities (IC_50_ = 0.72 and 0.85 µM, respectively) than **COE-16** (IC_50_ > 10 µM). This result implies that the presence of two -OMe groups in the chalcone oxime structure greatly enhances MAO-B inhibition as compared with single -OMe group. Moreover, replacing the *O*-aryl group with an *O*-chalcone group (viz. **COE-19** to **COE-23)** improved the MAO-B inhibitory activity. In the dimethoxy chalcone oxime series, a F in the acetophenone portion of the chalcone, **COE-22**, resulted in excellent MAO-B inhibitory activity (IC_50_ = 0.028 µM), as compared with unsubstituted (-H), **COE-19** and –OMe substituted **COE-20**. The single -OMe substituted chalcone oxime with -Me group in the acetophenone portion of the chalcone moiety of **COE-21** also showed significant difference in MAO-B inhibitory activity (IC_50_ = 0.036 µM). This result suggests that electronegative groups enhance MAO-B inhibitory activity. The acetophenone portion of chalcone containing the -OMe group in **COE-20** significantly increased MAO-B inhibitory activity, when compared to the benzaldehyde portion containing the same group in **COE-23** (IC _50_ = 0.35 vs. 0.15 µM). These SAR studies afford great scopes of opportunity to synthesize more potent chalcone-chalcone oxime hybrid molecules, i.e., COEs.

No SAR study could be performed with respect to MAO-A inhibitory activity. Only one compound, **COE-13**, showed significant MAO-A inhibitory activity with an IC_50_ value of 0.88 µM, which in itself suggested that the presence of an acetothiophene oxime group in the acetophenone portion might enhance the MAO-A inhibitory activity. Interestingly, **COE-13** also inhibited MAO-A and MAO-B.

Similarly, only compound **COE-22**, a chalcone-chalcone oxime ether with a –F substituent, inhibited AChE (IC_50_ = 4.39 µM). Thus, **COE-22** may be considered a MAO-B and AChE dual inhibitor for the treatment of neurodegenerative diseases.

### 2.4. Kinetics of MAO-B Inhibitions

Kinetic studies were performed on MAO-B inhibition by **COE-6** and **COE-22**. Lineweaver–Burk plots and secondary plots showed that **COE-6** and **COE-22** competitively inhibited MAO-B (Figure 3A,C) with K_i_ values of 0.0075 ± 0.00067 and 0.010 ± 0.0035 µM, respectively (Figure 3B,D). These results suggest that **COE-6** and **COE-22** are potent, selective, and competitive inhibitors of MAO-B.

### 2.5. Reversibility Studies

Reversibility studies were conducted on MAO-B inhibition by **COE-6** and **COE-22**. In these experiments, inhibitions of MAO-B by **COE-6** and **COE-22** were recovered from 19.7 (A_U_) to 81.1% (A_D_) and from 22.6 (A_U_) to 86.8% (A_D_), respectively (Figure 4), and these values were similar to those of the reversible reference inhibitor lazabemide (from 2.4 to 76.4%). However, inhibition by the irreversible reference inhibitor pargyline was only slightly recovered (from 3.7 to 10.4%). These experiments showed that inhibitions of MAO-B by **COE-6** and **COE-22** were recovered to the reversible reference level, which suggested that both are reversible inhibitors.

### 2.6. Computational Studies

Computational analyses were performed using QM-polarized docking and MM-GBSA calculations in order to investigate the binding modes of **COE-6** and **COE-22** towards MAO-A and MAO-B and with the purpose of clarifying the MAO-B selectivity of the two compounds. Table 2 reports calculated docking scores and the Δ***G*** binding values of the two compounds against MAO-A and MAO-B. In agreement with in vitro IC_50_ values, **COE-22** showed a better docking and MMGBSA scores for MAO-B compared to MAO-A. Conversely, **COE-6** did not show a significant gap in the docking score values between MAO-B and MAO-A.

Compound **COE-22** interacts with MAO-A and MAO-B with different binding modes (Figure 5). The residues of MAO-A involved in **COE-22** binding are Tyr62 and Lys218, which establish π and cation-π interactions, respectively, with the *para*-fluorine phenyl ring, Lys341, which engages a hydrogen bond with the carbonyl oxygen atom of the chalcone portion of **COE-22**, and Lys316, which forms a cation-π interaction with the *para*-methoxy phenyl ring, as shown on panel (a) of Figure 5. Notably, the distance between the *para*-fluorine phenyl ring and aromatic rings of the flavin adenine dinucleotide (FAD) molecule is ~11 Å. On the other hand, as shown on panel (b) of Figure 5, the residues of MAO-B involved in **COE-22** binding are similar to those hypothesized in previous studies [25], whereby the *para*-fluorine phenyl ring of **COE-22** is trapped within an aromatic cage made up of FAD, Tyr398, and Tyr435. Furthermore, the *para*-fluorine phenyl ring, the *para*-methoxy styryl, and the chalcone aromatic ring establish π−π interactions with Tyr398, Trp119, and Tyr236 (MAO-B selective residue), respectively. In addition, the carbonyl oxygen of the chalcone scaffold of **COE-22** forms a hydrogen bond with the thiol group of Cys172.

Docking analysis did not report meaningful differences for interactions of **COE-6** towards MAO-A and MAO-B. The *para*-fluorine phenyl ring of **COE-6** is involved in π−π interaction with Tyr407 of MAO-A, and it is trapped within an aromatic cage delimited by Tyr407, Tyr444, and FAD, unlike **COE-22**, as shown on panel (a) of Figure 6. When considering MAO-B, the *para*-fluorine phenyl ring of **COE-6** engages π−π interaction with Tyr435 and faces the aromatic cage that is formed by Tyr435, Tyr398, and FAD. Notably, the selective MAO-B residue Tyr326 establishes π−π interaction with the chalcone aromatic ring, and Cys172 makes a hydrogen bond with the carbonyl oxygen of the **COE-6** chalcone scaffold.

Docking studies carried out on **COE-6** and **COE-22** have proved that compound **COE-22** had the highest MAO-B affinity and appreciable selectivity. More specifically, **COE-22** interacts with MAO-A and MAO-B, but with different binding modes. In particular, in agreement with previous findings [24,25,26,27,28,29], the chalcone head of **COE-22** faces the FAD of MAO-B, whereas **COE-6** adopts similar poses for MAO-A and MAO-B, probably because of its smaller size. Interestingly, docking studies successfully explained at the molecular level the different experimental affinities of **COE-6** and **COE-22** for the two MAO isoforms. In particular, the gain in binding for MAO-B was mostly supported by the chance of forming π−π hydrophobic interaction with Tyr326. This is a key residue, which changed to I335 in MAO-A [49], capable of giving access to the binding pocket (for **COE-22** compound) and stabilizing the chalcone aromatic ring.

## 3. Materials and Methods

### 3.1. Enzyme Assays

Recombinant hMAO-A and hMAO-B activities were assayed while using kynuramine (0.06 mM) and benzylamine (0.3 mM) as substrates, respectively, as described previously [50]. The substrate concentrations were 1.7 × and 1.9 × K_m_, respectively (K_m_ = 0.036 and 0.16 mM, respectively). AChE activity was measured using Type VI-S from *Electrophorus electricus* in the presence of 0.5 mM 5,5’-dithiobis(2-nitrobenzoic acid) and 0.5 mM acetylthiocholine iodide, as described previously [51,52]. Enzymes and chemicals were purchased from Sigma–Aldrich (St. Louis, MO, USA).

### 3.2. Analysis of Enzyme Inhibitions and Kinetics

The inhibitory activities of the 24 COEs synthesized against MAO-A and MAO-B were first investigated at a concentration of 1.0 µM, and IC_50_ values were then determined. AChE inhibitory activities were also determined, except at a concentration of 10 µM. Time-dependent inhibitions and reversibilities were measured, and kinetic studies were performed on the most potent MAO-B inhibitors, i.e., **COE-6**, and **COE-22**, as previously described [53]. Kinetic experiments were carried out at five substrate and three inhibitor concentrations.

### 3.3. Analysis of Inhibitor Reversibility

The reversibilities of compounds **COE-6**, and **COE-22** were analyzed while using a dialysis method after preincubating with MAO-B for 30 min, as previously described [54]. Reversibilities were determined for **COE-6**, **COE-22**, lazabemide (a reversible MAO-B reference inhibitor), and pargyline (an irreversible MAO-B reference inhibitor) at ~2 × IC_50_ concentrations, i.e., 0.004, 0.050, 0.080, and 0.040 µM, respectively. Relative activities of undialyzed (A_U_) and dialyzed (A_D_) samples were used to determine the reversibilities.

### 3.4. Computational Studies

The three-dimensional (3D) structures of MAO-A (PDB ID: 2Z5X) and MAO-B (PDB ID: 2V5Z) were obtained from the Protein Data Bank. The protein preparation wizard available in the Schrödinger suite was used to optimize X-ray crystal structures [55,56]. MAO-A and MAO-B active sites contained nine and eight water molecules, respectively. The LigPrep tool was used to optimize ligand structures and generate possible tautomers and ionization states at physiological pH. Docking simulations were carried out using the QM polarized ligand docking protocol available from Schrödinger Suite. While retaining the rigidities of protein structures, QM polarized ligand docking allows for ligands with a certain degree of conformational flexibility. Centers of mass of X-ray cognate ligands of MAO-A and MAO-B structures were used as references for the cubic grid center.

The QM-polarized ligand docking protocol that was implemented in Glide was used with default options. This protocol uses three computational steps, that is: a) a standard precision (SP) initial docking using Glide; b) calculation of QM partial charges of the docked ligand based; and, c) a SP re-docking phase for each ligand pose when considering computed QM based charges.

A Molecular Mechanics/Generalized Born Surface Area (MM-GBSA) method was added to the workflow for the calculation of the binding free energies (Δ***G***) between protein and ligands in order to estimate ligand-binding affinities. Such a method is implemented in Prime available in the Schrodinger software 2018-2 (New York, NY, USA) [57]. Provided that Δ***E_MM_*** is the minimized energy of the ligand-protein complex, Δ***G_solv_*** is the solvation energy, and Δ***G_SA_*** is the binding energy of the surface area of compounds, with respect to MAO-A and MAO-B, Δ***G_bind_*** values were computed, as follows: (1)$$\Delta G_{bind}=\Delta E_{MM}+\Delta G_{solv}+\Delta G_{SA}$$

Obtained docking poses were minimized using Prime [57,58,59].

## 4. Conclusions

We evaluated the MAO-A, MAO-B, and AChE inhibitory activities of 24 previously synthesized chalcone oxime ethers (**COE-1**–**COE-24**). Most of the COEs exhibited significant and selective MAO-B inhibitory activity. Three compounds, viz., **COE-6**, **COE-7 (chalcone-ketoxime ethers),** and **COE-22 (chalcone-chalconeoxime ethers)**, potently inhibited MAO-B. However, only **COE-13** (**chalcone-ketoxime ethers)** significantly inhibited MAO-A and MAO-B. Notably, **COE-22** inhibited AChE well and potently inhibited MAO-B. Both lead MAO-B inhibitors, **COE-6** (**chalcone-ketoxime ethers)** and **COE-22**, contained a F substitution, which once again supports the notion that fluorine inclusion can have a profound effect on the biological activity. Reversibility and kinetics studies on **COE-6** and **COE-22** showed that both potently, selectively, reversibly, and competitively inhibited MAO-B. We hope this preliminary study on these novel COEs encourages medicinal chemists to further explore MAO inhibition and conduct biological activity studies.

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
