# Peer review of "Novel Class of Chalcone Oxime Ethers as Potent Monoamine Oxidase-B and Acetylcholinesterase Inhibitors"

_molecules, 2020, doi:10.3390/molecules25102356_

Round 1

Reviewer 1 Report

Easily available chalcone oxime ethers (COEs) by Pd-catalyzed C‒O cross-coupling were evaluated for inhibitory activities against monoamine oxidases (MAOs) and acetylcholinesterase (AChE). Most of synthesized COEs had potent and/or significant selective inhibitory effects on MAO-B. Some of them are slightly more potent than clorgyline, lazabemide, and pargyline (reference drugs). This paper contains syntheses of COEs, inhibitory activities against MAO-A, MAO-B, and AChE, SARs for inhibition studies, Kinetics of MAO-B inhibitions, Computational studies, others. This paper is worthy of publication in this Journal.

Author Response

We thank Reviewer for reviewing our manuscript. We would be really grateful to the Reviewer for accepting the manuscript for publication in journal “Molecules”.

Reviewer 2 Report

Molecules

Manuscript ID: Molecules-798615

Manuscript title: “Novel class of chalcone oxime ethers as potent monoamine oxidase-B and acetylcholinesterase inhibitors

By Oh, J.M. et al.

Dear Editor,

The paper “Novel class of chalcone oxime ethers as potent monoamine oxidase-B and acetylcholinesterase inhibitors” involves the evaluation of hybrid molecules of chalcone-oxime-ether class against monoamine oxidase-B and acetylcholinesterase enzymes, their possible mechanism of inhibition, SAR and molecular modelling studies.

The results are interesting and promising. In my opinion, the following questions should be to answer:

  1. In the page 2, the cited AChE inhibitors, such as, donepezil, rivastigmine and tacrine, could have their chemical structures shown for visualization of the functional groups.

         Similarly, to the MAO inhibitors cited.

  1. In the page 3, the oxime-ethers Cefmenoxime, Aztreonam, Ridogrel and Oxiconazole, also oxime-ethers, could have their chemical structures shown to better understanding the importance of oxime-ether group.
  2. In the page 4, 2.1. Synthesis of COEs, although their syntheses have been reported on the reference [46], I think that the author could report briefly about the principal method used and, the characteristic spectroscopic data, because in the reference [46] it was not done, that is, the structures have not been elucidated.
  3. In the page 5, line 155, the authors comment about similar to other chalcone derivative that the use of fluorine group led to increase the inhibitory activity of MAO-B, but this structure or name this chalcone derivative do not is cited.
  4. In the Conclusion, may be the authors could be to comment the structural difference between the compound COE-13, dual inhibitor and the COE-6 and COE-7 with selectivity to MAO-B. The compound CEO-22 have a different structure to comparation.

Author Response

We thank Reviewer for reviewing our manuscript. We would be really grateful to the Editor and Reviewer for accepting the manuscript with Minor Revisions.

The results are interesting and promising. In my opinion, the following questions should be to answer:

Reviewer’s Comment: 1

In the page 2, the cited AChE inhibitors, such as, donepezil, rivastigmine and tacrine, could have their chemical structures shown for visualization of the functional groups.  Similarly, to the MAO inhibitors cited.

Authors’ Response to comment 1: We thank Reviewer for valuable suggestions. Chemical structures have been incorporated in the revised manuscript as per suggestions.

Reviewer’s Comment: 2

In the page 3, the oxime-ethers Cefmenoxime, Aztreonam, Ridogrel and Oxiconazole, also oxime-ethers, could have their chemical structures shown to better understanding the importance of oxime-ether group.

Authors’ Response to comment 2: We thank Reviewer for valuable suggestions. Chemical structures have been incorporated in the revised manuscript as per suggestions.

Reviewer’s Comment: 3

In the page 4, 2.1. Synthesis of COEs, although their syntheses have been reported on the reference [46], I think that the author could report briefly about the principal method used and, the characteristic spectroscopic data, because in the reference [46] it was not done, that is, the structures have not been elucidated.

Authors’ Response to comment 3: We thank Reviewer for valuable suggestions. We are extremely sorry for the typo in the reference no. Actually, it was supposed to be 47. All details such as synthetic methods etc., are available with the ref. No. 47.

Reviewer’s Comment: 4

In the page 5, line 155, the authors comment about similar to other chalcone derivative that the use of fluorine group led to increase the inhibitory activity of MAO-B, but this structure or name this chalcone derivative do not is cited.

Authors’ Response to comment 4: We thank Reviewer for the comment. Actually, citing compound name(s) here in the manuscript is cumbersome as the compound name is too lengthy. Instead, we have clearly mentioned the compound code/no. We believe, now it is more clear, the thing you mentioned, in the revised manuscript.

Reviewer’s Comment: 5

In the Conclusion, may be the authors could be to comment the structural difference between the compound COE-13, dual inhibitor and the COE-6 and COE-7 with selectivity to MAO-B. The compound CEO-22 have a different structure to comparation.

Authors’ Response to comment 5: We thank Reviewer for the comment. The required corrections have been done in the revised manuscript. Thank you Very much for your valuable time and comments.